Molecular phylogeny and species delimitation of the freshwater prawn Macrobrachium pilimanus species group, with descriptions of three new species from Thailand

Siriwut Warut 1
Jeratthitikul Ekgachai 1
Panha Somsak 2 4
Chanabun Ratmanee 3
Sutcharit Chirasak chirasak.s@chula.ac.th 2
1 Department of Biology, Faculty of Science, Mahidol University , Bangkok , Thailand
2 Department of Biology, Faculty of Science, Chulalongkorn University , Bangkok , Thailand
3 Faculty of Agricultural Technology, Sakon Nakhon Rajabhat University , Sakhon Nakhon ,
4 Academy of Science, The Royal Society of Thailand, Bangkok , Thailand
Baird Donald
Electronic publication date: 2020 Nov 27
Publication date: 2020
Volume: 8
Electronic Location ID: e10137
Received 2020 Apr 6; Accepted 2020 Sep 18
Copyright: ©2020 Siriwut et al.
Copyright year: 2020
Copyright holder: Siriwut et al.
License: This is an open access article distributed under the terms of the Creative Commons Attribution License, which permits unrestricted use, distribution, reproduction and adaptation in any medium and for any purpose provided that it is properly attributed. For attribution, the original author(s), title, publication source (PeerJ) and either DOI or URL of the article must be cited.
License URL: https://creativecommons.org/licenses/by/4.0/

Keywords: Phylogeny, Freshwater, Species delimitation, Taxonomy and systematics, Thailand, Integrative taxonomy, New species

Funding: Center of Excellence on Biodiversity BDC-PG2-160012 Thailand Research Fund TRF-DPG6280001 This study was funded by the Center of Excellence on Biodiversity (BDC-PG2-160012) and the Thailand Research Fund (TRF-DPG6280001). The funders had no role in study design, data collection and analysis, decision to publish, or preparation of the manuscript.

==============================
Specific status and species boundaries of several freshwater prawns in the Macrobrachium pilimanus species group remain ambiguous, despite the taxonomic re-description of type materials and additional specimens collected to expand the boundaries of some species. In this study, the “pilimanus” species group of Macrobrachium sensu Johnson (1958) was studied using specimens collected from montane streams of Thailand. Molecular phylogenetic analyses based on sequences of three molecular markers (COI, 16S and 18S rRNA) were performed. The phylogenetic results agreed with morphological identifications, and indicated the presence of at least nine putative taxa. Of these, six morphospecies were recognised as M. malayanum, M. forcipatum, M. dienbienphuense, M. hirsutimanus, M. eriocheirum, and M. sirindhorn. Furthermore, three morphologically and genetically distinct linages were detected, and are described herein as M. naiyanetri Siriwut sp. nov., M. palmopilosum Siriwut sp. nov. and M. puberimanus Siriwut sp. nov. The taxonomic comparison indicated wide morphological variation in several species and suggested additional diagnostic characters that are suitable for use in species diagnoses, such as the shape and orientation of fingers, the rostrum form, and the presence or absence of velvet pubescence hairs and tuberculated spinulation on each telopodite of the second pereiopods. The “pilimanus” species group was portrayed as non-monophyletic in both ML and BI analyses. The genetic structure of different geographical populations in Thailand was detected in some widespread species. The species delimitation based on the four delimitation methods (BIN, ABGD, PTP and GMYC) suggested high genetic diversity of the “pilimanus” species group and placed the candidate members much higher than in previous designations based on traditional morphology. This finding suggests that further investigation of morphological and genetic diversity of Southeast Asian freshwater prawns in the genus Macrobrachium is still required to provide a comprehensive species list to guide efforts in conservation and resource management.

Introduction

Macrobrachium prawns have received particular attention worldwide because of their economic value and their use as model organisms for biogeographical study of evolutionary diversification (De Bruyn et al., 2014). Recently, evidence of high genetic diversity and species richness in some freshwater and terrestrial invertebrates in mainland Southeast Asia was revealed by integrating morphological and molecular systematic methods. Systematic studies of Asian shrimp and prawn species have been increasingly pursued due to evidence of unreported species and underestimation of genetic diversity (Bernardes et al., 2017; De Bruyn & Mather, 2007; De Mazancourt et al., 2019; Von Rintelen, Von Rintelen & Glaubrecht, 2007). New native species have been reported from several remote areas throughout both continental and insular Asia (Cai & Ng, 2002; Chong, 1989; Saengphan et al., 2018; Saengphan et al., 2019; Wowor & Short, 2007; Xuan, 2012).

In the past, Thai freshwater prawn and shrimp fauna were referred to in some taxonomic revisions among the oriental crustacean fauna (Holthuis, 1950; Holthuis, 1955; Johnson, 1963). Twenty-eight described species of freshwater prawns of the genus Macrobrachium Spence Bate, 1868 have been reported in Thailand (Cai, Naiyanetr & Ng, 2004; Naiyanetr, 2001; Naiyanetr, 2007; Saengphan et al., 2018; Saengphan et al., 2019). All Macrobrachium species in Thailand are found abundantly within two major riverine systems, namely the Chaophraya and Greater Mekong Basins, as reported by previous taxonomic studies (Cai & Ng, 2002; Hanamura et al., 2011). Cai, Naiyanetr & Ng (2004) reported that the M. pilimanus species group sensu Johnson (1960) consisted of 12 species: M. pilimanus (De Man, 1879), M. leptodactylus (De Man, 1892), M. hirsutimanus (Tiwari, 1952), M. dienbienphuense Dang and Nguyen, 1972, M. eriocheirum Dai, 1984 (currently treated as a synonym of M. dienbienphuense), M. ahkowi Chong and Khoo, 1987, M. gua Chong, 1989, M. forcipatum Ng, 1995, M. platycheles Ou and Yeo, 1995, M. pilosum Cai and Dai, 1999, M. amplimanus Cai and Dai, 1999, and M. sirindhorn Naiyanetr, 2001. Later, five new species were added to this species group: M. dalatense Xuan, 2003 from southern Vietnam, three species from Indonesia, namely M. urayang Wowor and Short, 2007, M. kelianense Wowor and Short, 2007, M. empulipke Wowor, 2010 and one troglobitic species, M. spelaeus Cai and Vidthayanon, 2016 from Thailand. The diagnostic characters for this group were critically debated due to complicated morphological variation. However, several species exhibit compatible patterns by having a short blade-like rostrum, cupped or slightly elongated carpus, swollen merus of the second pereiopods, and the presence of velvet setae on the telopodites of the second pereiopods (Cai, Naiyanetr & Ng, 2004; Chong, 1989; Holthuis, 1979; Johnson, 1960; Ng, 1994).

Several species in the “pilimanus” species group exhibit widespread distribution, such as M. dienbienphuense, M. amplimanus, M. hirsutimanus and M. forcipatum. In contrast, there are also some species reported to be endemic and limited to a narrow territory, including M. sirindhorn and M. spelaeus, which are restricted to areas in northern Thailand (Cai & Vidthayanon, 2016). Detailed information on the distribution range and type locality of all nominal taxa in “pilimanus” species group has been provided in Table S1. The limitation of using traditional taxonomic characters for species identification in the “pilimanus” group has been acknowledged, as several species exhibit similar morphological patterns and have few diagnostic characters (Holthuis, 1950; Johnson, 1960; Johnson, 1963; Ou & Yeo, 1995; Yeo, Cai & Ng, 1999). The diagnosis of nominal taxa has usually been based on a combination of quantitative and qualitative characters such as the proportion of rostrum, podomeres of second pereiopods and the presence and absence of pubescence on fingers, palm and merus of second pereiopods. Because of high morphological variation, the species diversity of M. pilimanus group has been debated (Cai, Naiyanetr & Ng, 2004; Cai & Liang, 1999; Hanamura et al., 2011; Holthuis, 1952; Johnson, 1960; Li et al., 2007; Wowor, 2010; Wowor & Short, 2007). Previously, the phylogenetic position refered by some M. pilimanus members also indicated the unclear relationship between congeneric species in genus Macrobrachium such as M. niphanae, M. yui and M. neglectum (Liu, Cai & Tzeng, 2007; Wowor et al., 2009).

Several taxonomic identifications of prawns in genus Macrobrachium were based on the combination of traditional morphology. The re-examination of type specimens and additional museum collections has been done in some Macrobrachium species (Cai, Naiyanetr & Ng, 2004; Cai & Shokita, 2006; Holthuis, 1952). The comprehensive distribution and taxonomic status of several species are questionable due to limited material available from different geographical areas and their scattered distribution ranges (Cai & Ng, 2002; Hanamura et al., 2011; Johnson, 1963). Although Thailand is located in the center of mainland Southeast-Asia, its freshwater fauna is likely under-reported, including Macrobrachium prawns in both major river basins. The lack of broad-scale specimen comparison and comprehensive data on geographical variation and genetic composition are of critical concern, given the obscure justification for their taxonomic boundaries (Castelin et al., 2017; Chen et al., 2015; Rossi & Mantelatto, 2013). As a result, classification and assignment of Macrobrachium species into a suitable species complex or species groups has generally been problematic (Johnson, 1960; Wowor & Ng, 2007; Wowor & Short, 2007).

Molecular systematics based on DNA barcoding regions and species delimitation coupled with DNA sequence variation has been widely used to screen for putative species identification in some highly diversified decapod groups (Bernardes et al., 2017; De Mazancourt et al., 2019; Venera-Pontón et al., 2020; Fujisawa & Barraclough, 2013). In this study, we integrate traditional taxonomic examination and molecular phylogeny using three molecular markers to delimit species boundaries and to illustrate the phylogenetic relationships within the “pilimanus” species group collected from Thailand, with further discussion of their distribution and phylogenetic position among mainland Southeast Asian species.

Methodology

Field collecting and specimen preparation

Prawn specimens were collected from riverine systems throughout Thailand. Field surveys were conducted to collect fresh specimens in some protected areas with permission from the Department of National Parks, Wildlife and Plant Conservation, Thailand (DNP 0907.4/14262). Some species previously described with the type locality in Thailand were re-collected and used as additional topotype material for species identity in morphological and molecular examinations. The live habitus specimens were photographed in order to document body colouration, and then euthanised by the two-step method following AVMA Guidelines for the Euthanasia of Animals (AVMA, 2013) before fixing in 95% ethanol for long-term preservation. Animal use in this study strictly followed the protocols approved by Chulalongkorn University (Protocol Review No. 1723018) and Mahidol University-Institute Animal Care and Use Committee (MU-IACUC) under approval number MU-IACUC 2018/004.

Collected prawn specimens were registered and housed at Chulalongkorn University Museum of Zoology, Bangkok, Thailand (CUMZ), and Mahidol University, Natural History Museum (MUNHM). Species identifications were made by comparison with previous taxonomic records of Macrobrachium prawns from Thailand and surrounding countries. Morphological characteristics of each species were observed by using stereo-microscope. Traditional and diagnostic characters for species identification were photographed with Cell’D imaging system. In addition, the fine detail of some morphological characters were illustrated by free-hand drawings to document their variation. For morphological variation analysis, constant characters were selected for study using classical landmark-based geometric morphometrics. The protocols used in this study followed Siriwut et al. (2015).

Species descriptions and technical terms used herein are based on previous taxonomic studies of Southeast Asian Macrobrachium species (Cai & Dai, 1999; Cai, Naiyanetr & Ng, 2004; Cai & Ng, 2002; Hanamura et al., 2011; Holthuis, 1950; Wowor & Short, 2007; Xuan, 2012). Abbreviations for terms used in the comparison table are as follows: Fin., fingers; Pal., palm; Carp., carpus; Mer., merus; Dt., teeth on dactylus; Pt., teeth on pollex. The rostrum teeth formula is the total number of dorsal teeth/total number of ventral teeth. Total body length (tl) used in the species description was measured from the end of the telson to the tip of the rostrum. Carapace length (cl) was measured from the dorso-posterior margin of the carapace to the end of the post-antennular margin of the carapace. Rostrum length (rl) was measured from the tip of the rostrum to the posterior-most rostrum tooth. All characters are reported in millimeters.

Nomenclatural acts

The electronic version of this article in portable document format (PDF) will represent a published work according to the International Commission on Zoological Nomenclature (ICZN), and hence the new names contained in the electronic version are effectively published under that Code from the electronic edition alone. This published work and the nomenclatural acts it contains have been registered in ZooBank, the online registration system for the ICZN. The ZooBank LSIDs (Life Science Identifiers) can be resolved and the associated information viewed through any standard web browser by appending the LSID to the prefix http://zoobank.org/. The LSID for this publication is: urn:lsid:zoobank.org:pub:F94C18CF-8E07-4D4B-94ED-4153854B237E. The online version of this work is archived and available from the following digital repositories: PeerJ, PubMed Central and CLOCKSS.

DNA extraction and PCR

All prawn samples used for molecular analysis in this study are listed in Table 1. Prawn tissue. The genomic DNA was extracted from abdominal muscle tissue by using Commercial DNA extraction kits (NucleoSpin Tissue kit; MACHEREY-NAGEL). The concentration of total genomic DNA was measured and visualised by gel electrophoresis. Three standard molecular loci for Macrobrachium were selected for phylogenetic study, including the barcode regions of mitochondrial cytochrome c oxidase subunit I (COI), 16S rRNA (16S), and nuclear 18S rRNA (18S). The criteria for DNA marker selection were (1) sequences of closely related taxa for sequence comparison are available in a public database such as GenBank and BOLD (Liu, Cai & Tzeng, 2007; Wowor et al., 2009), (2) marker is commonly used for phylogenetic tree reconstruction of genus Macrobrachium (Rossi et al., 2020; Saengphan et al., 2018; Saengphan et al., 2019) and (3) a sufficient amount of variation, conserved and parsimony informative sites for multi-locus phylogenetic study (Liu et al., 2017; Matzen da Silva et al., 2011; Pileggi & Mantelatto, 2010). The PCR primers used in amplification and sequencing are presented in Table 2. PCR reactions were incubated using T100™ thermal cycler (BIO-RAD) with gradient temperature function. The components of the PCR mixture followed Siriwut et al. (2015). Reaction conditions for each molecular locus were based on previous phylogenetic studies of shrimp and prawns (Pileggi & Mantelatto, 2010; Rossi & Mantelatto, 2013; Von Rintelen, Von Rintelen & Glaubrecht, 2007; Wowor et al., 2009). Successfully amplified PCR products were checked by using fluorescence-enhanced agarose gel electrophoresis.

The PEG precipitation method was used to purify the PCR products. The purified PCR products were sequenced at Bioneer Inc. (Korea). Raw sequences were aligned with libraries in GenBank using the BLASTn algorithm to verify the organism’s identity. Sequence configuration was done in Sequence Navigator (Parker, 1997). Sequence annotation and trimming were carried out in MEGA 7 (Kumar, Stecher & Tamura, 2016) using MUSCLE (Edgar, 2004). Sequence format was constructed using MEGA 7 and Mesquite (Maddison & Maddison, 2017). All newly obtained nucleotide sequences were deposited in the GenBank database under GenBank submission numbers MT235929-MT235968 for COI, MT248221-MT248260 for 16S, and MT248181-MT248220 for 18S (in Table 1).

Table 1 Locality with geographic coordinates and GenBank accession numbers for specimens used for molecular phylogenetic analyses.

Taxon	CUMZ-Voucher ID	Locality	Coordinates	GenBank accession NO.	
				COI	16S	18S	
Macrobrachium dienbienphuense Dang and Nguyen, 1972	CUMZ MP00020-M016	Khek River, Wangthong, Phitsanulok	16°52′26.9″N 100°38′25.8″E	MT235932	MT248224	MT248184	
	CUMZ MP00021-M027	Due Bridge, Yom River, Pong, Phayao	19°06′24.3″N 100°15′58.5″E	MT235934	MT248226	MT248186	
	CUMZ MP00022-M054	Kaeng Lamduan, Dom Pradit, Nam Yuen, Ubon Ratchathani	14°26′46.2″N 105°07′16.3″E	MT235943	MT248235	MT248195	
	CUMZ MP00023-M069	Hui Yang, Wang Sam Mo, Udon Thani	16°56′46.3″N 103°21′56.0″E	MT235945	MT248237	MT248197	
	CUMZ MP00024-M084	Bueng Sam Phan, Phetchabun	15°49′58.5″N 101°02′07.3″E	MT235947	MT248239	MT248199	
	CUMZ MP00025-M148	Dom Yai, Det Udom, Ubon Ratchathani	14°49′43.9″N 105°04′48.5″E	MT235963	MT248255	MT248215	
Macrobrachium eriocheirum Dai, 1984	CUMZ MP00026-M050	Khao Sok National Park, Phanom, Surat Thani	8°54′47.2″N 98°31′28.2″E	MT235939	MT248231	MT248191	
	CUMZ MP00027-M097	Xishuangbanna, Yunnan, China	21°56′01.5″N 101°15′04.7″E	MT235948	MT248240	MT248200	
	CUMZ MP00028-M098			MT235949	MT248241	MT248201	
	CUMZ MP00029-M138	Kaeng Sopha, Wang Thong, Phitsanulok	16°52′37.7″N 100°38′28.1″E	MT235961	MT248253	MT248213	
Macrobrachium forcipatum Ng, 1995	CUMZ MP00035-M130	Kathu Waterfall, Kathu, Phuket	7°55′56.1″N 98°19′23.5″E	MT235956	MT248248	MT248208	
	CUMZ MP00036-M130A			MT235957	MT248249	MT248209	
	CUMZ MP00037-M130B			MT235958	MT248250	MT248210	
Macrobrachium hirsutimanus (Tiwari, 1952)	CUMZ MP00030-M051	Petch Rimtarn Resort, Kaeng Krachan, Tayang, Phetchaburi	12°49′45.0″N 99°43′39.0″E	MT235940	MT248232	MT248192	
	CUMZ MP00031-M052	Wang Ta Krai Waterfall, Hin Tung, Mueang, Nakhon Nayok	14°19′17.6″N 101°18′22.1″E	MT235941	MT248233	MT248193	
	CUMZ MP00032-M053	Klong Soan Reservoir, Bo Rai, Trat	12°31′38.0″N 102°36′14.0″E	MT235942	MT248234	MT248194	
	CUMZ MP00033-M083	Chomphu Bridge, Noen Maprang, Phitsanulok	16°41′32.1″N 100°40′15.2″E	MT235946	MT248238	MT248198	
	CUMZ MP00034-M140	Hui Phra Prong, Kabin Buri, Prachin Buri	13°54′33.8″N 101°50′16.9″E	MT235962	MT248254	MT248214	
Macrobrachium malayanum (Roux, 1934)	CUMZ MP00038-M132	Roi Chan Phan Wang Waterfall, Wang Wiset, Trang	7°53′16.1″N 99°19′54.4″E	MT235959	MT248251	MT248211	
	CUMZ MP00039-M151			MT235964	MT248256	MT248216	
	CUMZ MP00040-M152			MT235965	MT248257	MT248217	
	CUMZ MP00041-M153			MT235966	MT248258	MT248218	
Macrobrachium naiyanetri sp. nov.	CUMZ MP00004-M102	Khao Banchob Waterfall, Makham, Chanthaburi	12°51′04.5″N 102°12′10.6″E	MT235951	MT248243	MT248203	
	CUMZ MP00002-M127	Hui Prik, Cha-wang, Nakhon Si Thammarat	8°35′41.2″N 99°27′55.6″E	MT235954	MT248246	MT248206	
	CUMZ MP00001-M128			MT235955	MT248247	MT248207	
	CUMZ MP00002-M154			MT235967	MT248259	MT248219	
	CUMZ MP00002-M155			MT235968	MT248260	MT248220	
Macrobrachium niphanae Shokita and Takeda, 1989	CUMZ MP00042-M023	Nam Ko, Lom Sak, Phetchabun	16°47′34.8″N 101°10′34.8″E	MT235933	MT248225	MT248185	
Macrobrachium neglectum (De Man, 1905)	CUMZ MP00044-M060	Klong Chalung, Mueang, Satun	6°43′13.3″N 100°03′49.6″E	MT235944	MT248236	MT248196	
Macrobrachium palmopilosum sp. nov.	CUMZ MP00010-M011	Mae Mang, Bo Kluea, Nan	19°08′12.7″N 101°09′01.2″E	MT235931	MT248223	MT248183	
	CUMZ MP00009-M030	Sob-Pue, Sa-Iap, Song, Phrae	18°40′20.6″N 100°13′26.1″E	MT235935	MT248227	MT248187	
	CUMZ MP00007-M031	Tat Man Waterfalls, Puea, Chiang Klang, Nan	19°17′11.9″N 100°47′20.0″E	MT235936	MT248228	MT248188	
Macrobrachium puberimanus sp. nov.	CUMZ MP00015-M049	Nam Soam, Noan Thong, Na Yung, Udon Thani	18°00′30.5″N 102°14′42.8″E	MT235938	MT248230	MT248190	
	CUMZ MP00012-M099	Wat Tha Khaek, Chiang Khan, Loei	17°54′17.7″N 101°40′58.4″E	MT235950	MT248242	MT248202	
	CUMZ MP00014-M121	Phu Ruea, Loei	17°26′11.0″N 101°19′30.8″E	MT235953	MT248245	MT248205	
Macrobrachium rosenbergii (De Man, 1879)	CUMZ MP00045-M115	Klong Phon Rang, Mueang, Ranong	9°53′12.5″N 98°38′00.6″E	MT235952	MT248244	MT248204	
Macrobrachium sirindhorn Naiyanetr, 2001	CUMZ MP00018-M009	Namtok Nam Min, Mae Lao, Chiang Kham, Phayao	19°26′46.2″N 100°26′26.3″E	MT235929	MT248221	MT248181	
	CUMZ MP00019-M010			MT235930	MT248222	MT248182	
Macrobrachium sintangense	CUMZ MP00043-M038	Bang Ban, Phra Nakhon Si Ayutthaya	14°22′20.5″N 100°28′55.8″E	MT235937	MT248229	MT248189	

Table 2 Details of primers used in this study (F, Forward; R, Reverse).

Gene	Primer name	Sequence (5′ to 3′)	Reference	
COI	LCO1490 (F)	GGT CAA CAA ATC ATA AAG ATA TTG G	Folmer et al. (1994)	
MacroNancy (R)	GCG GGT AGR ATT AAR ATR TAT ACT TC	This study	
16S	16Sa-L (F)	CGC CTG TTT ATC AAA AAC AT	Palumbi (1996)	
16Sbr-H2 (R)	CTC CGG TTT GAA CTC AGA TCA	Palumbi (1996)	
18S	18S-ai (F)	CCT GAG AAA CGG CTA CCA CAT C	DeSalle et al. (1992)	
18S-bi (R)	GAG TCT CGT TCG TTA TCG GA	Whiting et al. (1997)	

Phylogenetic reconstruction and species delimitation

For our phylogenetic study, the dataset of each partial gene was compiled from the newly amplified sequences from fresh material and available sequences from public databases (NCBI and BOLD). The number of sequences used per marker are as follows: 57 sequences for COI, 79 sequences for 16S, and 53 sequences for 18S. For the concatenated dataset, the number of sequences used for each marker was optimized in order to average individual sequence length of sample. Samples from public databases were included in the concatenated dataset when at least two of the three marker sequences were available. In total, 54 sequenced samples were used in phylogenetic tree reconstruction based on the concatenated dataset. A list of outgroups and other Macrobrachium taxa in this study is provided in Table S2.

Maximum likelihood (ML) and Bayesian inference (BI) methods were applied to reconstruct phylogenetic trees. The concatenated dataset of three genetic markers with the partitioned file for nucleotide substitution model fit was prepared using Kakusan 4 (Tanabe, 2007). The alternative substitution model for phylogenetic tree reconstruction was tested by using JModelTest v.1.7 (Posada, 2008). For ML analysis, RAxML 8.0.0v (Stamatakis, 2006) with default parameter set was used to reconstruct phylogenetic tree. The ML tree topology was confidential tested under 1,000 bootstrap replicates. Bayesian inference tree was sampled in MrBayes, ver. 3.2.6. (Ronquist et al., 2012). Markov chain Monte Carlo (MCMC) were configured to run for 10 million generations, and trees were saved each 500 generations. Twenty-five percent of tree samples were discarded under burn-in fragment parameter settings. The consensus tree was generated from a 50% majority rule. The annotation and illustration of clade and branch length were configured by FigTree (Rambaut, 2009). Node creditable values, bootstrap (ML) and posterior probablility (BI), are labelled on the clade based on the acceptance criteria as follow: bootstrap values exceed 70% (Larget & Simon, 1999) and posterior probabilities exceed 0.95 (Huelsenbeck & Hillis, 1993). A p-distance method was used to calculate the genetic distance of all gene fragments in MEGA 7. The nMDS plot of pairwise sequnce results was constructed for COI and 16S by using PAST program (Hammer, Harper & Ryan, 2001).

Species delimitation was performed using four standardised methods for automatic species delimitation to detect the Molecular Operational Taxonomic Units (MOTUs): automated barcode gap (ABGD by Puillandre et al., 2012), Bayesian implementation of Poisson Tree Processes model (bPTP by Zhang et al., 2013), the multi-rate Poisson Tree Processes (mPTP by Kapli et al., 2017) and the Generalized Mixed Yule Coalescent model (GMYC by Pons et al., 2006). Each gene dataset was tested separately as a single partition. For the COI dataset, the sequence analysis function in BOLD including BIN clustering was implemented to designate the possible putative species in sequence dataset. For the ABGD method, the intra-specific variation obtained from each molecular marker dataset was calculated in MEGA7 and the optimised barcode relative gap was calculated using the ABGD online server (http://wwwabi.snv.jussieu.fr/public/abgd/abgdweb.html). The PTP analysis was conducted under the Maximum likelihood algorithm using a web server (https://species.h-its.org/gmyc/; by Zhang et al., 2013). The best-scoring tree dataset was estimated under 95% confidence of statistical probability. In the GMYC method, the starting tree was randomly sampled and manually calculated under a suitable model for the construction of an ultra-metric tree using BEAST package v1.10.4 (Drummond & Rambaut, 2007; Suchard et al., 2018) or implemented in CIPRES (Miller, Pfeiffer & Schwartz, 2010). The maximum clade credibility tree from each gene analysis was summarised in TreeAnnotator v1.10.4 and was analysed under the GMYC species delimitation approach using an online server. The results of automatic delimitation methods were compared (1) with the morphological identification of genus Macrobrachium species based on their original descriptions and with recent taxonomic reviews of nominal taxa to match each clade under biological species and (2) with molecular phylogenetic partional analysis based on the three concatenated gene datasets.

Results

Phylogenetic relationship and species delimitation of Thai “pilimanus” species group

Thirty-nine sequences from three partial genes were successfully amplified and comparatively aligned. The sampling locality of each species is illustrated in Fig. 1. The annotation of each partial gene sequence is described in Table 3. The genetic distance of each mitochondrial DNA dataset (COI and 16S) and nuclear 18S dataset was calculated with 1,000 bootstrap replicates. The estimates of inter- and intra-specific variation of all representative taxa, are listed together with standard deviation in Table S3. Interspecific variation between members of the “pilimanus” species group found in Thailand was 9.8–23.3% for COI, 2.3–7.7% for 16S and 0.2–11% for 18S. Intraspecific variation was 0.45–8.36% for COI, 0-3.5% for 16S and 0-2.1% for 18S. Non-metric multidimensional scaling (nMDS) plots representing pairwise comparison of COI and 16S sequences used for single gene analysis (including sequences from NCBI and BOLD) were generated (see Fig. S1 and Tables S4–S5).

Figure 1 Sampling localities of Macrobrachium pilimanus group in this study.

Table 3 Sequence annotation and DNA substitution model of each partial molecular marker used in this study.

Molecular marker	Sequence length	Conservative site	Variable site	Parsimony-informative site	Substitution model for DNA evolution	
COI	678	428	250	228	TIM2+I+G	
16S	529	397	132	93	TPM3uf+G	
18S	678	428	250	228	TIM1+I	

The phylogenetic tree based on the concatenated dataset of three partial genes indicated the non-monophyletic relationship of genus Macrobrachium because two outgroups (Coralliocaris superba and Exopalaemon styliferus) were nested inside and represented polytomy (clade A in Fig. 2). This result was also found in single-locus phylogenetic analyses (see Figs. S2–S3). After being rooted by outgroups and additional “pilimanus” members (M. pilimanus and M. urayang), most “pilimanus” members except an OTU of M. urayang from Indonesia showed a monophyletic relationship and nested with M. niphanae, with values showing support in both BI and ML analyses (clade B). The monophyletic relationship of most “pilimanus” members was indicated in clade C, and they were separated into two linages. The clade D linage comprised six species: M. malayanum, M. naiyanetri sp. nov., M. forcipatum, M. sirindhorn, M. pilimanus, and M. palmopilosum sp. nov. The monophyly of M. malayanum was detected and it was positioned as a basal clade to other congeneric species within this linage. The phylogenetic tree also indicated the nesting of M. sirindhorn with two other species, namely M. pilimanus and M. palmopilosum sp. nov., although this clade was not supported by statistical tests. In clade F, specimens of M. forcipatum, M. naiyanetri sp. nov. and one sample referred to as M. aff. pilimanus formed a monophyletic group, with statistical support from both ML and BI analyses. Macrobrachium naiyanetri sp. nov. and M. aff. pilimanus formed a monophyletic group, while M. forcipatum was placed at the base of the clade. The monophyly of M. naiyanetri sp. nov. was further separated into two distinct geographical clades: a clade including samples from the southern peninsulaof Thailand plus M. aff. pilimanus from Khammouane, Laos, and a second clade of two samples from eastern Thailand.

Figure 2 Phylogenetic tree based on concatenated dataset of three molecular genes (COI, 16S and 18S rRNA), geographical distribution and morphological characteristics of second pereiopods of M. pilimanus species group.

(A) Phylogenetic tree (B) Morphological character of second pereiopods (C) Distribution area. Nodes of phylogenetic tree marked with empty circles indicate statistical support from both ML and BI (>70 bootstrap value and >0.97 posterior probability score); grey circles indicate statistical support from only one (either ML or BI); asterisk indicates the sample obtained from NCBI.

In clade E, Macrobrachium hirsutimanus, M. eriocheirum, M. dienbienphuense and M. puberimanus sp. nov. were nested as a monophyletic group with statistical support in both ML and BI. Within this clade, the phylogenetic positions of M. hirsutimanus and M. eriocheirum were uncertain due to low support of clade composition; however, the monophyletic relationship of representative OTUs was indicated consistently in ML and BI for both taxa. Clade G included two species with similar morphology, M. dienbienphuense and M. puberimanus sp. nov.; the monophyly of each species is questionable due to two sequences of M. dienbienphuense from the public database nested with M. puberimanus sp. nov. In the major clade of M. dienbienphuense, two genetically distinct subclades were found with statistical support.

Species delimitation of each partial sequence dataset indicated a different number of candidate taxa, and there was also variation by calculation approach (Fig. 2). The BIN clustering method in BOLD indicated 29 putative species for the COI dataset. The ABGD method indicated 19 species in COI, 19 species in 16S and 9 species in 18S. In the Bayesian Poisson Tree Process (bPTP), the clustering result indicated 20 species in COI, 19 species in 16S and 15 species in 18S. The multi-rate Poisson Tree Process (mPTP) indicated 14 species in COI, 2 species in 16S and 1 species in 18S. In the GMYC analysis, the clustering method indicated 18 species in COI, 21 species in 16S and 3 species in 18S, based on the ultrametric tree. The separation evidence (red box) detected eight taxa while lumping evidence (blue box) was found mainly in the clade of M. puberimanus sp. nov. and from two samples of M. dienbienphuense from the public database.

Systematic diversity of the “pilimanus” species group in Thailand

In this study, field collection and taxonomic identification of Thai Macrobrachium indicated nine morphological species, three of which are totally distinct from the others by both morphology and molecular delimitation. Six described species, namely M. hirsutimanus, M. eriocheirum, M. dienbienphuense, M. forcipatum, M. malayanum and M. sirindhorn were re-confirmed with previous taxonomic studies. The distribution of these six species mainly included montane tributary streams, while some species such as M. dienbienphuense also occupied larger rivers. The geographical distribution of “pilimanus” members is illustrated in Fig. 1. Based on this study and previous taxonomic records of Macrobrachium prawns in the “pilimanus” group, Thailand hosts eleven species. However, only the three new species found in this study will be described here, along with their phylogenetic placement, genetic relationship and geographical distribution.

Taxon names declaration: The proposed three new species herein are attributed to Warut Siriwut; thus, the authorship of these new taxon names should be cited as M. naiyanetri Siriwut in Siriwut et al., 2020, M. palmopilosum Siriwut in Siriwut et al., 2020 and M. puberimanus Siriwut in Siriwut et al., 2020.

Taxonomic account

Palaemonidae Rafinesque, 1815	
Macrobrachium Spence Bate, 1868	

Macrobrachium naiyanetri Siriwut sp. nov.

ZooBank ID: urn:lsid:zoobank.org:act:22EBCA17-2E29-4193-9D9E-87CABCD65D7D Figures 4A and 5

Figure 3 Results of species delimitation based on multiple approaches.

Abbreviations used on phylogenetic tree are as follow: Morpho, morphological identification; Phylo, phylogenetic analysis; BIN, BIN clustering in BOLD; ABGD, automated barcode gap; bPTP, Bayesian Poisson tree processes; mPTP, multi-rate Poisson Tree Processes; GMYC, Generalized Mixed Yule Coalescent model. Box colours indicate the split (red) and lumped (blue) species recognized by each species delimitation method. Grey boxes indicate non-monophyly in phylogenetic analysis and missing sequences from dataset in each delimitation method; asterisk indicates the sample obtained from NCBI.

Figure 4 Live habitus specimens of three new Macrobrachium species in the M. pilimanus group from Thailand.

(A) Macrobrachium naiyanetri sp. nov. (B) Macrobrachium palmopilosum sp. nov. (C) Macrobrachium puberimanus sp. nov.

Type locality. A large and shallow stream with large gravels at Hui Prik, Cha-wang District, Nakhon Si Thammarat Province, Thailand.

Type examined. Holotype: CUMZ MP00001, one male spm. from Hui Prik, Cha-wang District, Nakhon Si Thammarat Province (M128 in molecular analysis). Paratype: CUMZ MP00002, four male spms from the same locality as holotype (M127, M154 and M155). CUMZ MP00003, nineteen male and nine female spms from the same locality as holotype.

Additional material. CUMZ MP00004, two male spms from Khao Banchob Waterfall, Makham District, Chanthaburi Province (M102). CUMZ MP00005, one male spm. from Klong Rattaphum, Rattaphum District, Songkhla Province (M134). CUMZ MP00006, twenty-six male and nine ovigerous female spms from Klong Krabiead, Hui Prik, Cha-wang District, Nakhon Si Thammarat Province.

Diagnosis. Rostrum short and striate distally, not reach beyond the end of second telopodite of antennular peduncle. Rostral formula: 8-14/2-4 teeth. Small spinulation presents on anterolateral margin of carapace. Epistome trilobed. Second pereiopods slightly longer than body lenght, similar in shape, unequal in size. Second pereiopods with long setae,present on finger, palm, anterior inner part of carpus and merus. 10–18 teeth on figers. Carpus elongated or slightly cupped, shorter than fingers, palm and merus. All telopodites of second pereiopods covered with spinules. Thoracic sternites; T4 with postero submedial plate; T5 with transverse plate with median process. Second and third abdominal sternites with moderate triangular median process. Preanal carina present. Telson slightly short and stouth, surface glabrously, with long plumose seta and posterior projection with two long inner and two short outer spines. Uropods glabrous; uropodal diaeresis with inner moveable spine, equal to outer angle. Developed eggs large, approximate diameter 0.7 mm, ovoid.

Composite description (type specimens in parentheses). A medium-sized Macrobrachium species, tl 30.6–54.2 mm (41.5 mm in holotype), with pale or brownish body colouration (Fig. 4A).

Rostrum (Figs. 5C and 5D). Anteriorly striate and angled downward distally, rl 7.3–11.4 mm (10.8 mm in holotype) cl 6.7–13.0 mm (13.0 mm in holotype), and reaching not beyond the end of antennular peduncle. Dorsal part of rostrum with 8-14 (14 in holotype) teeth in total, 2–7 (6 in holotype) teeth present in postorbital area. Area with postorbital teeth covering nearly half of carapace length. Ventral part of rostrum with 2–4 (3) teeth, located about half-way distally.

Figure 5 Morphological characters of Macrobrachium naiyanetri sp. nov. (A-G, I from holotype, H from paratype; CUMZ MP00003).

(A) Lateral view (B) Uropods (C) Carapace (D) Rostrum form and teeth (E) Major second pereiopod (F) Teeth on finger of major second pereiopod (G) Major second pereiopod length (H–I) Second pereiopods in female (J) Third pereiopod.

Cephalon. Eye well developed. Ocular beak moderately developed, without laterally expanded tip. Postantennular carapace margin rounded. Cornea osculum longer than stalk. Antennular peduncle longer than wide, lateral carina well developed, dorsal carina without sinuous. Antero-lateral part of carapace with antennal (one side without antennal spine in holotype). Small hepatic spines present lower than orbital angle; located behind ; branchiostegal suture present starting from hepatic spine to carapace margin. Spinulation present on ventro-lateral part of carapace (Fig. 5C). Epistome trilobed. Scaphocerite with margin concave laterally, distolateral tooth minutes and not reaching the end of lamella. Third maxilliped not reaching beyond antennal peduncle.

First pereiopods. Long and slender, reaching beyond the end of scaphocerite. Fingers about as long as palm; carpus longer than merus. Carpus, merus and ischium covered with small spinules. Scattered setae present on all segments but dense on finger and ischium.

Second pereiopods. Robust and longer than body length, similar in both shape and form; carpus of both major and minor second pereiopods extending beyond the end of scaphocerite.

Major second pereiopod (Figs. 5E and 5G). Spinulation present on all segments except fingers and palm. Fingers, palm, inner margins of carpus covered by fine setae. Dense, fine setae present on proximal part of finger. Merus with setae in some specimens. Fingers slender and longer than palm (17.6: 11.1 mm), finger bending with gap and tips crossed when closed in males. Dactylus with 10–18 (15) prominent teeth, basal teeth larger than distal teeth, pollex with 10–18 (12) teeth (Fig. 5F). Teeth sub-equally distributed and concealed by long velvety setae, without oblique carina distally. Upper and lower margins of palm slightly expanded. Carpus elongated, shorter than merus (7.6: 11.8 mm in holotype). Merus equal to palm (11.8 mm in holotype). Ischium tapered, shorter than merus.

Minor second pereiopod (Fig. 5H–I). Similar in form but shorter than major cheliped, spinulation present on all segments except fingers and palm. Fine setae densely covering proximal part of fingers and palm. Dactylus with 6–18 small teeth, pollex with 8–15 small teeth. Teeth sub-equally distributed, only half of finger length, concealed by long, fine setae. Oblique carina present on distal part, about one-third of finger length. Carpus elongated, shorter than merus. Merus subcylindrical and equal to palm. Ischium tapered, shorter than merus.

Third pereiopods (Fig. 5J). Long and slender, propodus extending to the end of scaphocerite. Small spinulation present on all segments except ischium. A fine seta present on all segments. Dactylus short (2.1 mm in holotype) and curved, with dorsolateral setae; ventral carina well developed. Propodus long (4.6 mm in holotype), with 6–8 (7) ventral pairs of spines distributed along length of propodus; carpus shorter than propodus (3.1 mm in holotype), with dorsal projection on distal part. Merus longer than carpus (5.6 mm in holotype). Ischium shorter than merus and carpus (2.8 mm in holotype).

Fourth and fifth pereiopods. Dactylus extending to the end of scaphocerite. Spinulation present on all segments except ischium. Scattered fine setae present on all segments. Propodus with 5–7 pairs of ventral spines distributed along its length, 2 corner spines with grouped setae on distal part. Carpus shorter than propodus and merus, with dorsal projection on distal part. Ischium shorter than merus and carpus.

Thoracic sternum. T4 without median process. T5 with transverse plate without median process. T8 with posteromedial lobes in males.

Abdomen. Usually smooth, with tiny spinules on pleural margins of first and second abdominal segments. All abdominal sternites with transverse ridge. Second and third abdominal sternites with moderate triangular median process, subsequent segment without process. The sixth sternite with median obtuse process. Preanal carina present, with group of small setae at tip in males.

Telson (Fig. 5B). slightly short and stouth (5.9 mm in holotype), lateral margins straight. Cluster of setae present on antero-median part. Dorsal surface with 2 pairs of dorsal spines. Projection present on posterior margin, with two spines and plumose setae on each side, inner pair of posterior spines longer than outer spines.

Uropods (Fig. 5B). Uropodal diaeresis with inner moveable spine, equal to outer angle. Exopod longer than broad (5.5: 2.5 mm in holotype) and not reaching the end of endopods.

Etymology. The specific name naiyanetri is given in honor of Professor Phaibul Naiyanetr from Chulalongkorn University for his extensive contributions to the knowledge of crustacean fauna in Thailand.

Size. Males slightly larger than females; the largest male recorded being 54.2 mm tl, 13.0 mm cl; the largest female 39.8 mm tl, 9.5 mm cl and egg size is 0.7 mm in diameter.

Distribution. Most populations are restricted to the southern part of Thailand; however, one specimen collected from Chantaburi Province extends its recorded distribution range to include the eastern part of Thailand.

Remarks. Macrobrachium naiyanetri sp. nov. resembles other members of the “pilimanus” species group by having densely tufted setae on second pereiopods. The phylogenetic tree suggests the position of this new species as nesting with M. forcipatum. However, the distinguishing characteristics of M. naiyanetri sp. nov. used to separate it from the other congener species in southern Thailand (e.g., M. forcipatum, M. malayanum and M. hirsutimanus) are the carpus of the second major pereiopods that exhibit a slight cup-shape, the presence of dense stiff setae on the antero-inferior part of merus, and fingers of the second pereiopods being longer than palms. Moreover, the postorbital area contains more rostrum teeth (4–7 vs. 3–5 in M. forcipatum; 3–4 in M. malayanum; 3–5 in M. hirsutimanus). The adult size of M. naiyanetri sp. nov. is significantly larger and longer than the others (tl). The dactylus contains 12–13 prominent teeth (vs. 13–14 in M. forcipatum; 4–6 in M. malayanum; 15 in M. hirsutimanus). The size of major and minor second pereiopods is distinctly large in male specimens (vs. not distinct in other species). The carpus of the second pereiopod is slightly cupped (vs. cupped and stout in other species). The major second pereiopod in males is as long as tl. In addition, the species delimitation methods suggest two distinct evolutionary lineages of M. naiyanetri sp. nov. samples; the first lineage is composed of specimens from the western part of Khao Luang Range, whereas the second lineage contains two samples from the eastern part of Khao Luang Range (Songkhla Province) and from Chantaburi Province in eastern Thailand. Further investigation of population structure between these two distinct lineages is necessary to test whether or not this is the result of allopatric speciation.

Macrobrachium palmopilosum Siriwut sp. nov.

ZooBank ID: urn:lsid:zoobank.org:act:8065628A-4EDF-49EF-BA5D-91588F53D284 Figures 4B and 6

Figure 6 Morphological characters of Macrobrachium palmopilosum sp. nov. (A-G, I from holotype, H from paratype; CUMZ MP00008).

(A) Lateral view (B) Uropods (C) Carapace (D) Rostrum form and teeth (E) Major second pereiopod (F) Teeth on finger of major second pereiopod (G) Major second pereiopod length (H–I) Second pereiopods in female (J) Third pereiopod.

Type locality. A small and shallow stream with sand and gravel at Tat Man Waterfalls, Puea Sub-district, Chiang Klang District, Nan Province, Thailand.

Type examined. Holotype: CUMZ MP00007, one male spm. from Tat Man Waterfalls, Puea Sub-district, Chiang Klang District, Nan Province (M031). Paratype: CUMZ MP00008, twenty-one male and twenty-seven female spms from the same locality as holotype.

Additional material. CUMZ MP00009, six male and two female spms from Sob-Pue, Sa-Iap Sub-district, Song District, Phrae Province (M030). CUMZ MP00010, twelve male and ten female spms from Mae Mang, Bo Kluea District, Nan Province (M011). CUMZ MP00011, one male spm. from Ban Pha Lak, Mueang District, Nan Province.

Diagnosis. Rostrum short, anteriorly striate and upward distally, not reaching to the end of second telopodite of antennular peduncle. Rostral formula: 10–12/2–3 teeth. Anterolateral margin of carapace with small spines. Epistome bilobed. The robust pair of second pereiopod longer than body length similar in shape, unequal in size. Densed andtufted setae present on both side of second pereiopods. Anterior part of carpus with setae. Fingers with 10–12 teeth. Carpus stoutand cupped, shorter than fingers, palm and merus. oSmall spinule present in posteriorpart of palm, entirely in carpus and merus. Thoracic sternites: T4 with posterior submedial plate; T5–T7 with transverse plate without median process; T8 with contiguous posteromedially anterior lobes, without median process. First to third abdominal sternites with moderate triangular median process. Preanal carina present. Telson moderately long, with scaterred plumose setae on dorsal surface. Two pairs of spines present. Posterior projection present with two long inner and short outer spines. Uropodal diaeresis spine shorter than outer angle. Egg size 1.3 mm in diameter.

Composite description (type specimens in parentheses). A medium-sized Macrobrachium species, tl 25.6–77.8 mm (57.3 mm in holotype), with pale or greenish-brown body colouration (Fig. 4B).

Rostrum (Figs. 6C and 6D). Anteriorly striate and turned upward distally, rl 4.1–16.7 mm (11.7 in holotype) cl 5.9–20.4 mm (16.5 mm in holotype), and reaching not beyond the end of second segment of antennular peduncle. Dorsal part of rostrum with 10–12 (12 in holotype) teeth in total, 4–6 (5) teeth present in postorbital area. Area with postorbital teeth covers one-third of carapace length. Ventral part of rostrum with 2-3 (3) teeth, located about half-way to distal end.

Cephalon. Eye well developed. Ocular beak moderately developed, without laterally expanded tip. Postantennular carapace margin rounded. Cornea osculum shorter than stalk. Antennular peduncle longer than wide, lateral carina well developed, dorsal carina without sinuous. Antero-lateral part of carapace with antennal spine. Small hepatic spines present lower than orbital angle and antennal spine. Branchiostegal suture starting from hepatic spine to carapace margin. A few scaterred spinules present on ventro-lateral part of carapace and branchiostegal regions of carapace (Fig. 6C). Ocular beak moderately developed, without laterally expanded tip. Epistome slightly bilobed. Scaphocerite with margin concave laterally, distolateral tooth minutes and not reaching the end of lamella. Third maxilliped not reaching beyond antennal peduncle.

First pereiopods. Long and slender, reaching beyond the end of scaphocerite. Fingers about as long as palm; carpus as long as merus. Small spinules present only on merus and ischium. Scattered setae present on all segments but dense area on distal part of finger and on entire ischium. The proximal part between palm and carpus with group of small setae.

Second pereiopods. Robust and longer than body length, similar in form; carpus of both major and minor second pereiopods extending beyond the end of scaphocerite.

Major second pereiopod (Figs. 6E and 6G). Spinulation present in all segments except fingers and anterior part of palm. Fingers, palm, inner margins of carpus covered by tufted setae. Merus without setae. Fingers subcylindrical, shorter than palm in length (13.8: 15.9 mm.), closed fingers with gap and crossing distally. Dactylus with 10–12 (10) prominent teeth, basal teeth smaller than middle teeth, pollex with 10–11 (11) teeth (Fig. 6F). Teeth sub-equally distributed and concealed by long tufted setae, without oblique carina distally. Upper and lower margins of palm slightly expanded. Carpus cup-shaped, shorter than merus (7.1: 13.9 mm). Merus slightly shorter than palm (13.9: 15.9 mm), stout and inflated laterally. Ischium tapered, shorter than merus.

Minor second pereiopod (Fig. 6H–I). Similar in form to major cheliped but smaller in size, spinulation present on all segments except fingers and anterior part of palm. Tufted setae covering fingers, palm and anterior part of carpus. Dactylus with 6–8 (6) small teeth, pollex with 7–8 (8) small teeth. Teeth distributed only on basal half of finger length, concealed by long, fine setae. Oblique carina present on distal part, about half of finger length. Carpus cup shaped, shorter than merus. Merus subcylindrical and as long as palm. Ischium tapered, shorter than merus.

Third pereiopods (Fig. 6J). Dactylus short (1.9 mm) and curved distally, with lateral short seta and ventral carina well developed. Propodus extending to the end of scaphocerite. Small spinulation present on all segments except ischium. A fine seta present on all segments. Propodus longer than dactylus (6.5: 1.9 mm), with 5–6 (6) ventral pairs of spines distributed along length of propodus. Carpus shorter than propodus (3.6 mm), with dorsal projection on distal part. Merus longer than carpus (6.5 mm). Ischium shorter than merus and carpus (3.3 mm).

Fourth and fifth pereiopods. Dactylus extending to the end of scaphocerite. Spinulation present on all segments except ischium. Scattered fine setae present on all segments. Propodus with 5–6 pairs of ventral spines distributed along length of propodus. Propodus of fifth pereiopods with group of setae on distolateral part. Carpus shorter than propodus and merus, with dorsal projection on distal part. Ischium shorter than merus and carpus.

Thoracic sternum. T4-T8 with transverse plate without median process. T8 with posteromedial lobes in males.

Abdomen. Usually smooth, with tiny spinules on pleural margin of first to third abdominal segments in some specimens. All abdominal sternites with transverse ridge. First to third abdominal sternites with moderate triangular median process. Fifth sternite without median obtuse process. Preanal carina present, without small setae in males.

Telson (Fig. 6B). Moderately long (6.6 mm) Dorsal surface with 2 pairs of spines. Cluster of setae present on antero-median part. Projection present on posterior margin, with two spines and plumose setae on each side. The inner pair of posterior spines longer than outer spines.

Uropods (Fig. 6B). Uropodal diaeresis with inner moveable spine, shorter than outer angle. Exopod longer than broad (7.4: 4.3 mm) and not reaching the end of endopods.

Etymology. The specific name “palmopilosum” is a compound Latin word with “palma” meaning palm of the hand and “pilosus” meaning hairy. This name refers to the tuft of hairs present on the palms of both second pereiopods.

Size. Males showing distinctly larger body size than females; the largest male recorded being 77.8 mm tl, 20.4 mm cl; the largest female 48.2 mm tl, 12.0 mm cl and egg size is 1.3 mm in diameter.

Distribution. Their distribution is restricted to the northern part of Thailand, Nan Province.

Remarks. The population of this new species is dominant in the Nan River Basin, especially living in clear, cool mountain streams. The colouration of this species varied from light pale to dark brownish; the banding pattern on the dorso-lateral part of tergum was observed in some individuals. Macrobrachium palmopilosum sp. nov. shares several characteristics with M. eriocheirum, M. amplimanus and M. hirsutimanus. The character distinguishing M. palmopilosum sp. nov. from M. eriocheirum and M. hirsutimanus is the presence of tufted setae on the palms of the second pereiopods. Macrobrachium hirsutimanus and M. eriocheirum exhibited tufted setae only on the anterior half of the palms, whereas M. palmopilosum sp. nov. had setae present over the entire surface of palms. Moreover, the spinulation on the anteromarginal surface of the carapace is always present in M. palmopilosum sp. nov. (absent in M. eriocheirum and M. hirsutimanus). The epistome of M. palmopilosum sp. nov. is slightly bilobed (trilobed in M. eriocheirum and M. hirsutimanus). The number of prominent teeth on fingers of M. palmopilosum sp. nov. is 6-12, whereas M. hirsutimanus has 12-20 teeth and M. eriocheirum has 12-15 teeth. Macrobrachium palmopilosum sp. nov. differs from M. amplimanus by having more rostrum teeth on the postorbital area (4–6 vs. 2–4), slightly smaller number of finger teeth on second pereiopods (10–12 vs. 11–15), the spinulation on palm surface of second pereiopods (present vs. absent), the length of fingers shorter than palm (vs. longer or as long as palm), and closed fingers with a gap (vs. without gap). The morphological comparisons of M. palmopilosum sp. nov. and other species are presented in Table 4.

Table 4 Morphological comparison of three new species and the closely related species in the M. pilimanus group recorded from Thailand.

Characters	Species	
	M. niyanetrisp.nov.	M. palmopilosumsp.nov.	M. puberimanussp.nov.	M. amplimanus*	M. dienbienphuense	M. hirsutimanus*	M. eriocheirum	
Rostrum teeth	8-14/2-4	10-12/2-3	12 − 15∕3	9 − 12∕2	8 − 14∕1 − 3	10/2	10 − 13∕2 − 3	
Rostrum reaching end of antenular peduncle	Not reaching to the end	Not reaching to the end	Reaching to the end	Not reaching to the end	Reaching to the end	Not reaching to the end	Not reaching to the end	
Spinule on margin of carapace	present	present	absent	present	present/absent	absent	absent?	
Epistome	trilobed	bilobed	trilobed	trilobed	trilobed	bilobed	trilobed	
Tuberculation/spine on palm surface of second pereiopods	absent	present	present	present?	present	absent	absent	
Length of male second pereiopods	unequal	unequal	unequal	unequal	unequal	unequal	unequal	
Segment of major second pereiopod	Fing.>Pal. Pal>Carp. Carp<Mer. Pal. =Mer.	Fing.<Pal. Pal.>Carp. Carp.<Mer. Pal. ≤Mer.	Fing.>Pal. Pal>Carp. Carp.<Mer. Pal. =Mer.	Fing.=Pal. Pal>Carp. Carp.<Mer. Pal. ≥ Mer.	Fing.>Pal. Pal>Carp. Carp.<Mer. Pal. ≥Mer.	Fing.<Pal. Pal ≥Carp. Carp.<Mer. Pal. ≥Mer.	Fing. ≥Pal. Pal>Carp. Carp.<Mer. Pal. =Mer.	
Carpus shape	Slightly elongate/cup	cup	elongate	cup	elongate	cup	cup	
Teeth on dactylus (Dt) and pollex (Pt)	Dt:10-18 Pt:10-18	Dt:10-12 Pt:10-11	Dt:11-16 Pt:10-14	Dt:13 Pt:13	Dt:20-32 Pt:20-32	Dt:15 Pt:15	Dt:12-15 Pt:12-15	
Gap in closed fingers	gapping	gapping	gapping	Not gapping	Not gapping	Slightly gapping	Slightly gapping	
Moveable spine on uropodal diaraesis	Equally to outer angle	Shorter than outer angle	Shorter than outer angle	Shorter than outer angle	Shorter than outer angle	Shorter than outer angle	Shorter than outer angle	
Notes.

‘*’ indicates data were retrieved from original description and “?” were data deficiency.

The results of phylogenetic tree construction suggested that M. palmopilosum sp. nov. is closely related to M. naiyanetri sp. nov., as supported by all statistical tests. Macrobrachium palmopilosum sp. nov. shows distinctive differences from M. naiyanetri sp. nov. by the stout cup shaped carpus of the major second pereiopods (vs. slightly elongated carpus in M. naiyanetri sp. nov.), the lack of setae on antero-inferior part of the merus of second pereiopods (vs. with dense setae on merus in M. naiyanetri sp. nov.), the inflated form of merus in M. palmopilosum sp. nov. (vs. subcylindrical in M. naiyanetri sp. nov.).

Tiwari (1952) described M. hirsutimanus based on specimens from northern Thailand (Doi Chuang) and later the type locality was replaced by the neotype designation (Nan Province; in Cai, Naiyanetr & Ng, 2004). This taxonomic treatment advocates that the distribution of M. hirsutimanus coexists with M. palmopilosum sp. nov. In this study, the coexistence of these two species of prawns was confirmed in the Nan River Basin.

Macrobrachium puberimanus Siriwut sp. nov.

ZooBank ID: urn:lsid:zoobank.org:act:EE26BC6C-07F6-4C94-8B80-6F736B11F91A Figures 4C and 7

Figure 7 Morphological characters of Macrobrachium puberimanus sp. nov. (A-G, I from holotype, H from CUMZ MP00015).

(A) Lateral view (B) Uropods (C) Carapace (D) Rostrum form and teeth (E) Major second pereiopod (F) Teeth on finger of major second pereiopod (G) Major second pereiopod length (H) Second pereiopods in female (I) Third pereiopod.

Type locality. Mekong River at Wat Tha Khaek, Chiang Khan Sub-district, Chiang Khan District, Loei Province

Type examined. Holotype: CUMZ MP00012, one male spm. from Wat Tha Khaek, Chiang Khan Sub-district, Chiang Khan District, Loei Province (M099). Paratype: CUMZ MP00013, two male spms from the same locality as holotype.

Additional material. CUMZ MP00014, one male spm. from Phu Ruea District, Loei Province (M121). CUMZ MP00015, four male and twelve female spms from Nam Soam, Noan Thong Sub-district, Na Yung District, Udon Thani Province (M049). CUMZ MP00016, four male spms from Mekong River, Chiang Khan Sub-district, Chiang Khan District, Loei Province. CUMZ MP00017, one male spm. from Mekong River, Pak Chom District, Loei Province.

Diagnosis. Rostrum moderately long, anteriorly striate and angled upward distally, reaching beyond the end of second segment of antennular peduncle. Rostral formula: 12–15/3 teeth. Carapace with small spinulation on anterolateral margin. Epistome trilobed. Second pereiopods strong and robust, shorter than body length, similar in shape and unequal in size. Long-tufted setae present on finger and palm of second pereiopods. Fingers of major second pereiopod with 11–16 teeth. Closded fingers with gap and crossing distally. Carpus elongated, shorter than palm. Spinulation present on dorso-inferior surface of palm, carpus, merus and ischium. Minor second pereiopod slight with tiny spines on each segment. Thoracic sternites: T4 with posterior submedial plate; T4–T7 with basolateral median plate without median notch; male T8 with posteromedially anterior lobes. Male and female without posteriorly medial process on T8. First to third abdominal sternites with moderate triangular median process. Preanal carina present. Telson moderately long, with long plumose setae on proximal part. Telson surface with two pairs of dorsal spines, terminal projection with two long inner and short outer spines. Uropodal diaeresis spine shorter than outer angle.

Rostrum short, anteriorly striate and upward distally, not reaching to the end of second telopodite of antennular peduncle. Rostral formula: 10-12/2-3 teeth. Anterolateral margin of carapace with small spines. Epistome bilobed. The robust pair of second pereiopod similar in shape, unequal in size. Densed andtufted setae present on both side of second pereiopods. Anterior part of carpus with setae. Fingers with 10-12 teeth. Carpus stoutand cupped, shorter than fingers, palm and merus. oSmall spinule present in posteriorpart of palm, entirely in carpus and merus. Thoracic sternites: T4 with posterior submedial plate; T5-T7 with transverse plate without median process; T8 with contiguous posteromedially anterior lobes, without median process. First to third abdominal sternites with moderate triangular median process. Preanal carina present. Telson moderately long, with scaterred plumose setae on dorsal surface. Two pairs of spines present. Posterior projection present with two long inner and short outer spines. Uropodal diaeresis spine shorter than outer angle. Egg size 1.3 mm in diameter.

Composite description (type specimens in parentheses). A medium-sized Macrobrachium species, tl 33.6–60.2 mm (60.2 mm in holotype), with pale or brownish-green body colouration (Fig. 4C).

Rostrum (Figs. 7C and 7D). Anteriorly striate and angled upward distally, rl 7.4–12.7 mm (12.7 mm in holotype), cl 6.6–17.0 mm (17.0 mm in holotype), and reaching beyond the end second segment of antennular peduncle. Dorsal part of rostrum with 12-15 (13) teeth in total, 5-6 (5) teeth present in postorbital area. Area bearing postorbital teeth covering one-fourth of carapace length. Ventral part of rostrum with 3 (3) teeth, located about half-way to distal end.

Cephalon. Eye well developed. Postantennular carapace margin rounded. Cornea osculum as long as stalk. Antennular peduncle longer than wide, lateral carina slightly concave, dorsal carina not sinuous. Sharp antennal and hepatic spines present at lower orbital angle; hepatic spine smaller, situated behind and below antennal spine; branchiostegal suture running from hepatic spine to anterior margin of carapace. Carapace without spinulation on ventro-lateral part and branchiostegal regions (Fig. 7C). Ocular beak moderately developed, without laterally expanded tip. Epistome trilobed. Scaphocerite, lateral margin slightly concave, distolateral tooth not reaching the end of lamella. Third maxilliped reaching beyond antennal peduncle and covering 75–80% of length of scaphocerite; ultimate slightly shorter than penultimate.

First pereiopods. Long and slender, reaching beyond the end of scaphocerite. Fingers about as long as palm; carpus as long as merus. Few setae scattered on all segments but dense on distal part of finger and on lower margin of ischium. Proximal part between palm and carpus without small setae.

Second pereiopods. Robust and slightly shorter than body length, similar in form but differing in size. Carpus of major second pereiopods extending beyond the end of scaphocerite.

Major second pereiopod (Figs. 7E and 7G). Spinulation present on dorso-inferior surface of palm, carpus, merus and ischium. Fingers, palm, inferior margins of carpus covered with few tufted setae. Merus without tufted setae anteriorly. Fingers sharp and subcylindrical, longer than palm in length (19.7: 15.3 mm), closed fingers with gap and crossing distally. Dactylus with 11–16 (16) prominent teeth, basal teeth slightly smaller than distal teeth, pollex with 10–14 (14) teeth (Fig. 7F). Teeth sub-equally distributed and concealed by long tufted setae, with oblique carina distally, about 15–20% of finger length. Upper and lower margins of palm not expanded. Carpus slightly elongated, shorter than merus (9.2: 16.6 mm). Merus subcylindrical, as long as palm or shorter (16.6 vs 15.3 mm). Ischium tapered, shorter than merus.

Minor second pereiopod (Fig. 7G). Short and smaller than major cheliped, spinulation absent in all segments. Few tufted setae covering fingers and palm. Dactylus with 6–8 (6) small teeth, pollex with 5–11 (7) small teeth. Teeth distributed only on basal half of finger length, concealed by fine setae. Oblique carina present on distal two-thirds of finger length. Carpus elongated, shorter than merus. Merus subcylindrical and as long as palm. Ischium tapered, shorter than merus.

Third pereiopods (Fig. 7I). Long and slender; propodus extending to the end of scaphocerite. Small spinulation absent in all segments. A fine seta present on all segments. Dactylus short and curved (2.2 mm), with dorsolateral setae on distal part, ventral carina well developed. Propodus longer than dactylus (6.5: 1.8 mm), with 5–7 ventral pairs of spines distributed along length of propodus. Carpus shorter than propodus (3.5 mm), with dorsal projection on distal part. Merus longer than carpus (8.6 mm). Ischium shorter than merus (3.2 mm).

Fourth and fifth pereiopods. Dactylus extending to the end of scaphocerite. Spinulation absent on all segments. Few fine setae present, scattered on all segments. Propodus with 5-6 pairs of ventral spines distributed along length of propodus. Propodus of fifth pereiopods with group of setae on distolateral part. Carpus shorter than propodus and merus, with dorsal projection on distal part. Ischium shorter than merus.

Thoracic sternum. T4–T7 with transverse plate without median process. T8 with posteromedial lobes in males.

Abdomen. Smooth, without small spinules on pleural margin of abdominal segments. All abdominal sternites with transverse ridge. First to third abdominal sternites with moderate triangular median process. Fifth sternite with median obtuse process. Preanal carina present, without small setae in males.

Telson (Fig. 7B). Moderately long (6.7 mm) andstraight. Dorsal surface with 2 pairs of spines. Cluster of setae present on antero-median part. Projection present on posterior margin, with two spines and plumose setae on each side. The inner pair of posterior spines longer than outer spines.

Uropods (Fig. 7B). Uropodal diaeresis with inner moveable spine, shorter than outer angle (Fig. 7B). Exopod longer than broad (8.0: 3.7 mm) and not reaching beyond the end of endopods.

Etymology. The specific name “puberimanus” is derived from the compound Latin words “puberis” for downy and “manus” for hand. It alludes to the long-tufted hairs on the second pereiopods.

Size. Males with larger body size than females; the largest male recorded being tl 60.0 mm, cl 17.0 mm; the largest female tl 28.9 mm, cl 8.8 mm; egg size is 1.7 mm in diameter.

Distribution. Recent populations are restricted to the northeastern part of Thailand and possibly occur in the Mekong River and its tributaries in Laos.

Remarks. This species is distributed commonly in tributaries of the middle Mekong River Basin in northeastern Thailand. The molecular phylogeny and morphological characters of M. puberimanus sp. nov. indicated close resemblance to M. dienbienphuense, which is commonly found in the Mekong River Basin, including Thailand, Laos, Cambodia (?), Vietnam, and also southern China (Hanamura et al., 2011). The characters distinguishing M. puberimanus sp. nov. from M. dienbienphuense are the number of finger teeth on the cutting edge of the major second pereiopod (11–16 vs. 18–32), spinulation on the anterior margin of carapace (absent vs. present), the spinulation on merus surface (sparse vs. abundant), and the slightly elongated carpus of second pereiopods (vs. highly elongated carpus).

Recently, a cavern-dwelling species was found from the central part of Thailand, namely M. spelaeus by Cai & Vidthayanon (2016). The morphological characters indicate similarity with M. dienbienphuense in several aspects except for the form of the anterior rostrum, the reduced eye, the bilobed epistome and the second pereiopod being as long as the body. In this study, M. puberimanus sp. nov. shows morphological differences from the latter species by having less elongated carpus, distal part of rostrum not upturned, and merus of second pereiopods with less spinulation. The distribution of M. puberimanus sp. nov. seems associated with the open riverine system of the Mekong River Basin, whereas the distribution of M. spelaeus is restricted to subterranean limestone systems in the central part of Thailand. Two additional species that resemble M. dienbienphuense and are co-distributed in the Mekong River Basin are M. amplimanus and M. eriocheirum. Hanamura et al. (2011) reviewed the morphological characters of these two species based on specimens from Laos and provided additional 16S rRNA sequences for molecular phylogenetic analysis. In this study, the 16S rRNA sequences of M. puberimanus sp. nov. were totally separated from Laotian M. amplimanus sequences, whereas M. eriocheirum from Laos nested within M. puberimanus sp. nov. samples (see Fig. S2 in supplement). However, the Laotian M. eriocheirum differs from M. eriocheirum sensu Dai (1984) in some aspects such as the number of dorsal and ventral rostrum teeth (9-12/2-3 vs. 11-14/2-3 in Laotian specimens) and the number of teeth on fingers of second pereiopods (10 teeth vs. 11-17). For this reason, the samples called M. eriocheirum in Hanamura et al. (2011) herein are excluded from this study; either they are M. puberimanus sp. nov. or a separate species.

Morphological diagnosis and shape variation

Key to mainland SE-Asian species in “pilimanus” species group (modified from Cai, Naiyanetr & Ng, 2004)

1. (a) A rudimentary appendix interna present on the first male pleopod……M. dalatense

(b) A rudimentary appendix interna absent on the first male pleopod…………………2

2. (a) Merus of second pereiopods with pubescence…………………………………….5

(b) Merus of second pereiopods without pubescence…………………………………3

3. (a) Rostrum short and convex distally, second pereiopods with fingers shorter than palm……………………..…………………………………………………………4

(b) Rostrum short and straight distally, second pereiopods with fingers shorter than palm…………………………………………………………M. naiyanetri sp. nov.

4. (a) 17 small teeth on fingers of second pereiopod…………………………..M. pilosum

(b) 8-10 blunt teeth on fingers of second pereiopod……………………..M. sirindhorn

5. (a) Tuberculation present on palm surface of second pereiopods…………………….6

(b) Tuberculation absent on palm surface of second pereiopods…………………….10

6. (a) Elongated carpus of major second pereiopod……………………………………7

(b) Cupped carpus of major second pereiopod ……………………………………8

7. (a) Cutting edges of fingers of second pereiopod with 23-32 teeth, closed fingers without gap………………………………….. …………………………M. dienbienphuense

(b) Cutting edges of fingers of second pereiopod with 11-16 teeth, closed fingers with gap……………………………………..…………………M. puberimanus sp. nov.

8. (a) Carapace margin without spinulation ……………………………………….. 9

(b) Carapace margin with spinulation………………..……M. palmopilosum sp. nov.

9. (a) Rostrum teeth arrangement 4+7/2, cutting edges of fingers of second pereiopod 7-10…………………………………………………………..………M. forcipatum

(b) Rostrum teeth arrangement 6+7/2, cutting edges of fingers of second pereiopod 11-12……………………………………………………………………M. pilimanus

10. (a) Epistome bilobed………………………………………………M. hirsutimanus

(b) Epistome trilobed…………………………………….. ……………………11

11. (a) Velvet pubescence on fingers and palm of second pereiopods……M. eriocheirum

(b) Densely tufted pubescence on fingers and palm of second pereiopods …………………..…………………………………………..……M. amplimanus

Using geometric-morphometric measurements, shape variation among species was detected by ten classical landmarks on the rostrum and carapace (Fig. S4). Canonical variates analysis (CVA) displayed sharp variation among nine species in the “pilimanus” group (Fig. S5). Shape variation between M. palmopilosom-M. naiyanetri was detected in both Procrustate and Mahalanobis distance analyses. In terms of Mahalanobis distance, the comparison of shape measurements resulted in seven paired species with statistical support (P < 0.0001). A summary of Procrustate and Mahalanobis distance analyses is given in Table S7.

Discussion

Phylogenetic relationship of “pilimanus” species group members in mainland Southeast Asia

The monophyletic status of genus Macrobrachium is still questionable based on samples of “pilimanus” members and other Macrobrachium species used in this study. The insertion of outgroups, Coralliocaris superba and Exopalaemon styliferus, within a clade of genus Macrobrachium contradicted previous phylogenetic studies indicating the separation of these two genera from genus Macrobrachium (Saengphan et al., 2018; Wowor et al., 2009). The selection of outgroup rooting is critical in phylogenetic analysis in order to clarify the evolutionary history of Macrobrachium species as indicated in previous reports (Murphy & Austin, 2005; Wowor et al., 2009). However, like this study, broad scale sampling of decapod phylogeny has shown that within family Palaemonidae, the genus Macrobrachium can be either nested with other genera such as Cryhiop, Exopalaemon and Palaemon or inserted within another closely related genus i.e., Leptopalaemon (Bracken, De Grave & Felder, 2009).

Molecular phylogenetic analysis of three partial gene datasets indicated at least ten different evolutionary lineages in the “pilimanus” species group. The two major clades (Fig. 1) are usually found in mainland Southeast Asia tributaries; clade E consists of M. dienbienphuenses + M. puberimanus sp. nov. + M. eriocheirum + M. hirsutimanus, and clade D consists of M. forcipatum + M. naiyanetri sp. nov. + M. palmopilosum sp. nov. + M. malayanum + M. pilimanus + M. sirindohrn. Macrobrachium sirindhorn is further grouped with M. pilimanus. The morphological characters of M . sirindhorn are quite unique and distinct from the congeners in this species group by having tufted setae on carpus and merus (except M. naiyanetri sp. nov., which has a group of stiff setae on the inner side of carpus and merus) and the distal downward pattern of rostrum. The distribution range of M. sirindhorn is questionable due to scattered records from northern Thailand and Laos. Another species that presents similar characters to M. sirindhorn is M. pilosum Cai and Dai, 1999 from southern China (Yunnan) and possibly northern Vietnam. Without genetic data of M. pilosum, we would keep these two as distinct valid species.

Macrobrachium species in clade E exhibited sympatric distribution in several river systems in north-central and eastern Thailand. Macrobrachium dienbienphuense and M. puberimanus sp. nov. exhibited elongated carpus of second pereiopods. However, the gap and slender shape of pollex and dactylus, and fewer spinules on the merus of second pereiopods are morphologically diagnostic characters of M. puberimanus sp. nov. The collected sample of M. puberimanus sp. nov. included a smaller number of individuals than for M. dienbienphuense in every locality. This finding might suggest a low population density of M. puberimanus sp. nov. in its natural habitat.

In clade E, Macrobrachium hirsutimanus and M. eriocheirum are morphologically distinct from each other by having incomplete covering of velvet setae on the palms of second pereiopods; however, they typically co-exist in the Chaophraya River Basin of Thailand. Macrobrachium eriocheirum Dai, 1984 was originally described from Yunnan and recently treated as a synonym of M. dienbienphuense. In this study, specimens from Yunnan (M97-M98) indicated genetic compatibility with Thai samples by forming a monophyletic relationship. This finding might suggest the validity of M. eriocheirum as mentioned by previous taxonomic studies (Cai, Naiyanetr & Ng, 2004; Hanamura et al., 2011). In addition, the southern population of M. eriocheirum was collected from peninsular Thailand, extending the known distribution range of this species. A taxonomic review of M. hirsutimanus has been made and the neotype designation of this species was described using specimens from Nan Province, northern Thailand (Cai, Naiyanetr & Ng, 2004). In this study, we sampled the northern riverine areas including the Nan River Basin to obtain a representative collection of specimens. The phylogenetic tree indicated a monophyletic group among molecular samples of M. hirsutimanus. However, there is another species from North-Central Thailand that is similar in morphological characters to M. hirsutimanus, namely M. spelaeus. The distribution of M. spelaeus is restricted to the underground freshwater system in a limestone cave; however, there may be some connection with the Nan River.

The mainland Southeast Asia “pilimanus” species group includes species from the southern peninsula of Thailand and a part of the Mekong River Basin (clade C in Fig. 2). Macrobrachium forcipatum and M. malayanum exhibited small body length and short second pereiopods. Previously, two samples of M. malayanum were reported from Narathiwat, Southern Thailand (Cai, Naiyanetr & Ng, 2004). In this study, two genetically diverse lineages of M. malayanum were found from the same locality. This might suggest the endemism of M. malayanum, which has restricted distribution in some natural habitats in the southern part of Thailand. The two new species in this study, M. naiyanetri sp. nov. and M. palmopilosum sp. nov. are grouped with M. pilimanus and M. sirindhorn; however, the phylogenetic relationship between these two species is questionable due to low statistical node support. Geographical differentiation in samples of M. naiyanetri sp. nov. was detected, and some species delimitation methods (ABGD, PTP and GMYC with mitochondrial loci) suggested the possibility of cryptic speciation for the two geographically different populations.

Species boundary of “pilimanus” species group designated by morphological and molecular delimitation methods

The Macrobrachium pilimanus group was initially proposed by Johnson (1960) and by the morphological concept, they shown the high morphologically complex group (Johnson, 1963). The high phenotypical variation was previously observed by Holthuis (1950), who reported morphologically complex forms of a single species, Macrobrachium pilimanus. In M. pilimanus sensu stricto (Johnson, 1963), the features used to diagnose this species from the two conspecific species (M. leptodactylus and M. malayanum) are the short fingers without a gap, inner edge of carpus of second pereiopods convex, and short rostrum. Geographical variation was detected in Javanian M. pilimanus and Bornean M. leptodactylus by either having a small number of teeth on rostrum or slightly different pattern of second pereiopods. The specimens were later re-examined, and found to be either the same species (Javanian M. pilimanus = M. leptodactylus; Ou & Yeo, 1995) or two distinct species (Bornean M. leptodactylus = M. urayang; Wowor & Short, 2007). The type re-examination of some members in the “pilimanus” species group show inappropriate species boundaries applied previously, or even the co-existence of unknown species within the type series, such as the specimens of M. malayanum in Johnson (1960) and Johnson (1963) was found morphologically differ from type of M. malayanum by Roux (1934) (see taxonomic treatment in Chong & Khoo, 1987b), while in another case, four paralectotype specimens of M. leptodactylus were found to be a distinct species, M. empulipke (see Wowor, 2010). In this study, a geometric morphometric examination of the M. pilimanus group was conducted for the first time. The use of measurable characters for species delimitation has been successful in several taxa with statistical confirmation (see Figs. S4–S5 and Tables S6-S7). Broad sampling, optimal specimens of each of the “pilimanus” members, and other landmark methods are required in further study. This study result would be relieved the alternative approach to delimit the species boundary of M. pilimanus species group under morphological species concept.

The phylogenetic relationships within the M. pilimanus group have never been specifically investigated in order to verify the group’s phylogenetic position and taxonomic validity. However, some members, including M. pilimanus, M. dienbienphuense, M. eriocheirum, and M. amplimanus were previously included in several large-scale phylogenetic studies of genus Macrobrachium or higher taxa (Bracken et al., 2010; Jose & Harikrishnan, 2019; Liu, Cai & Tzeng, 2007; Pileggi & Mantelatto, 2010; Wowor et al., 2009). A taxonomic review of some “pilimanus” members based on molecular delimitation was done by Hanamura et al. (2011), and morphological identification was supported by the 16S sequences to confirm the biological species concept. Using single genetic markers, the combination of available sequences from previous literature and newly amplified sequences from this study indicated unresolved phylogenetic relationships. The effect of long-branch attraction caused by gap insertion and short sequence length has been found in several database sequences. However, the M. pilimanus group shows a close evolutionary relationship with M. niphanae, another common species group found in mainland SE-Asia, based on the concatenated dataset.

In this study, the molecular delimitation using the optimum sequence dataset agreed well with traditional morphological classification despite the few diagnostic characters that have been observed in some nominal species. The monophyletic clade of each representative taxon detected from the concatenated dataset, including the three new species, was subsequently confirmed by automatic delimitation approaches based on single gene datasets (Fig. 2). The BIN algorithm reflects the highest number of putative species in the COI dataset. According to the barcode gap threshold, over-estimation might be caused by genetic divergence of the dataset used, including the previously deposited sequences. The point of caution for BIN delimitation results in this study seems to be obscurity on the species identification concept in several deposited sequences, especially in M. dienbienphuense. The genetic divergence among samples named M. dienbienphuense raises warning (max intraspecific divergence higher than Nearest-neighbor species; see Table S8) in barcode gap analysis. The BIN discordance also detects the non-compatibility of sequence divergence and BIN assignment which agree with the barcode gap threshold. In the case of mPTP and bPTP, the delimitation results showed moderate support for the designation of three new species found in this study. The clustering reassigned some “pilimanus” members to be a single species, as inferred in the 16S dataset under bPTP and mPTP. In the case of GMYC, the delimitation using COI and 16S agreed with morphological identification and phylogenetic clade composition. In this study, all delimitation methods also presented the warning of cryptic speciation in samples assigned as M. malayanum. Unsurprisingly, low success in using 18S rRNA sequences was found with ABGD, PTP and GMYC; clustering lumped members of the “pilimanus” group into one to three putative species. Furthermore, the COI barcoding region seems to provide the fine resolution required for genus Macrobrachium. This suggestion has also been reported in recent studies of DNA barcode application on marine decapods, including Macrobrachium prawns (Hernawati et al., 2020; Matzen da Silva et al., 2011).

The integrative approaches applied herein resolve the problems of morphological concordance among “pilimanus” members. However, the species boundaries delimited by traditional identification seem to be carefully interpreted when abundant samples were used for comparison according to geographical variation. A combination of morphology and molecular taxonomy approaches is recommended for future species delimitation in the M. pilimanus group for the following reasons: first, the molecular operational taxonomic unit (MOTUs) is helpful to accelerate the sample clustering process under traditional identification despite morphological complexity; second, the phylogenetic species concept can be used to force the species assignment and taxonomic validity when diagnostic characters of paired species are shown as unclear; third, molecular taxonomy can provide supporting evidence of cryptic speciation.

Species diversity and distribution of the Thai “pilimanus” species group

Recent taxonomic reviews of Thai Macrobrachium species included nine species belonging to the “pilimanus” species group: M. eriocheirum, M. hirsutimanus, M. dienbienphuense, M. forcipatum, M. amplimanus, M. malayanum, M. sirindhorn and M. spelaeus (Cai, Naiyanetr & Ng, 2004; Cai & Vidthayanon, 2016). In this study, seven previously recognised species were studied along with three new species that morphological and molecular datasets suggest should be grouped in the “pilimanus” species group. However, there are two nominated species in the Thai freshwater fauna that were not included in this study: M. amplimanus and M. spelaeus. The distribution of M. amplimanus has been reported from Thailand in four provinces, namely Chiang Mai, Loei, Kanchanaburi and Narathiwat (Cai, Naiyanetr & Ng, 2004); it is also present in Laos (Hanamura et al., 2011). Cai, Naiyanetr & Ng (2004) reported that the characteristics of M. amplimanus are very similar to M. forcipatum and M. hirsutimanus in several aspects. The distinguishing features that can be used to identify M. amplimanus are the short rostrum, stoutly-inflated second pereiopods, and the number of rostrum teeth. The collected specimens from the Mekong River in this study indicated only two morphological species: M. dienbienphuense and M. puberimanus sp. nov. However, the available 16S DNA sequences in GenBank of M. amplimanus used in Hanamura et al. (2011) were initially combined with the 16S dataset in this study (Table S2 in appendix). The results indicated that the Laotian sequences of M. amplimanus sensu Hanamura et al. (2011) resembled species within the M. eriocheirum clade. To confirm the true taxonomic identity of these samples, new analyses using a combination of molecular markers are required due to high variation of sites detected in the 16S rRNA gene.

Macrobrachium spelaeus, the only Thai cavern species, was reported from Phra Wang Dang Cave in Phitsanulok Province (Cai & Vidthayanon, 2016). This species resembles M. dienbienphuense in morphology by having bilobed epistome, convex anterior rostrum, reduced eye, and by the length of the major second pereiopod being as long as the body. In this study, we could not find any specimens that resembled the morphology of M. spelaeus from central or northern Thailand. Moreover, fresh materials for DNA analysis of this species is limited, and gaining access to the exact location of the type locality is difficult due to conservation efforts. However, the samples from neighboring rivers and small streams indicated two species that possibly co-exist with this species: M. eriocheirum and M. dienbienphuense.

Previously, the study of freshwater prawn genus Macrobrachium mainly focused on the commercial species due to their economic value both globally and at a local scale (New & Nair, 2012). Recently, two newly named species, M. suphanense and M. chinatense were described from freshwater tributaries in central Thailand (Saengphan et al., 2018; Saengphan et al., 2019). There is also some genetic evidence of Thai Macrobrachium species exhibiting distinct geographical populations (Khanarnpai, Thaewnon-ngiw & Kongim, 2019; Saengphan et al., 2018). In total, thirty-one Macrobrachium species have been reported from Thailand, including the three new species found in this study. These findings suggest that the species diversity of freshwater fauna in Thailand has been under-reported and needs more attention. Furthermore, several native species of the genus Macrobrachium in Thailand and adjacent areas are of critical concern due to disturbance by anthropogenic activity, especially taxa in the “pilimanus” species group. The habitat preference of these prawn species is usually small streams or river systems connected to mountainous territory, which recently have been impacted by tourism and plantation development. The water quality and current flow of several riverine systems in mainland Southeast Asia are monitored under several environmental and ecological programs (Dudgeon, 2000; Hughes, 2017; Todd, Ong & Chou, 2010). Changes of the tributary system may cause the ecosystem to collapse by the disruption in species composition and loss of native freshwater fauna (Fukushima et al., 2014). However, the baseline data on biology, taxonomy and ecology are still insufficient. For this reason, further studies on biology, systematics and ecology of native Macrobrachium species are still required, especially in the context of biogeographical distribution related to migration, and river tributaries and their flows (De Bruyn, Wilson & Mather, 2004; Wowor et al., 2009). The integration of recent novel methods such as molecular phylogeny, species distribution modeling and ecological monitoring methods would be beneficial for database implementation in conservation management of freshwater prawns at both local and regional scales (De Grave, Cai & Anker, 2008; De Grave et al., 2015; Michael, 1988).

Conclusion

In this study, the integrative approach provided additional three new species of M. pilimanus members found in montain stream of Thailand. The species delimitation method related to biological and phylogenetic species concepts provided an alternative scheme for the justification of species boundary in this Macrobrachium species group. The geographical variation, refered both in molecular and morphological characteristics was documented in some species of M. pilimanus and would suggest the differences of dispersal abilities among congeneric species. The phylogenetic relationship among M. pilimanus members still be controversy due to non-monophyly but at least the mainland SE-Asian species united as monophyletic clade. The genetic variation based on this study and deposited samples suggests the possible cryptic fauna in Macrobrachium prawns from mainland SE-Asia where the massive network river basin was recognized. The distribution area of mainland M. pilimanus indicated the trend of species composition and abundant related to water flows from two basins; Chao Phraya and Mekong.

Supplemental Information

Figure S1 nMDS plot of pairwise sequence comparison from COI and 16S datasets

Click here for additional data file.

Figure S2 Phylogenetic tree based on single-gene reconstruction of COI, 16S and 18S sequences from public database and this study

Click here for additional data file.

Figure S3 Phylogenetic tree based on single-gene reconstruction of COI, 16S and 18S sequences from public database and this study under optimized sampling criteria

Click here for additional data file.

Figure S4 Landmark point and wired-frame illustration indicating shape variation

Click here for additional data file.

Figure S5 CVA clustering results for nine species in the “pilimanus” group

Click here for additional data file.

Table S1 The type locality and distribution range of members in M. pilimanus group

Click here for additional data file.

Table S2 GenBank accession numbers of additional sequences including outgroup taxa used in this study; n/a is data not provided in reference literature or database

Click here for additional data file.

Table S3 The interspecific variation of DNA sequence in COI gene

The blue text indicates standard deviation.

Click here for additional data file.

Table S4 nMDS sample annotation base on COI data

Click here for additional data file.

Table S5 nMDS sample annotation base on 16S data

Click here for additional data file.

Table S6 The result of Procrustes ANOVA analyses

Click here for additional data file.

Table S7 Mahalanobis and Procrustes distances of shape variation from pairwise-comparison of species classifiers in CVA method

Click here for additional data file.

Table S8 Result of barcode gap analysis

Click here for additional data file.

The authors would like to give sincere thanks to members of the Animal Systematics Research Unit, Chulalongkorn University (ASRU) and Animal Systematics and Molecular Ecology laboratory, Mahidol University (ASME) for kind support during field collecting and data analysis. Cordial thanks for accommodation and technical support during this study are given to all staff in the Department of Biology, Faculty of Science, Mahidol University. Field surveys in many restricted areas were supported by staff of the Department of National Parks, Wildlife and Plant Conservation. The authors would like to express our grateful thanks to reviewers for their constructive comments that improved the manuscript.

Additional Information and Declarations

Competing Interests

Author Contributions

Ethics

Field Study Permissions

DNA Deposition

Data Availability

New Species Registration

The authors declare there are no competing interests.

Warut Siriwut conceived and designed the experiments, performed the experiments, analyzed the data, prepared figures and/or tables, authored or reviewed drafts of the paper, and approved the final draft.

Ekgachai Jeratthitikul and Chirasak Sutcharit conceived and designed the experiments, performed the experiments, analyzed the data, authored or reviewed drafts of the paper, and approved the final draft.

Somsak Panha and Ratmanee Chanabun conceived and designed the experiments, authored or reviewed drafts of the paper, and approved the final draft.

The following information was supplied relating to ethical approvals (i.e., approving body and any reference numbers):

Chulalongkorn University (Protocol Review No. 1723018) and Mahidol University-Institute Animal Care and Use Committee (MU-IACUC; MU-IACUC 2018/004) approved the study.

The following information was supplied relating to field study approvals (i.e., approving body and any reference numbers):

The Department of National Parks, Wildlife and Plant Conservation, Thailand provided permission (DNP 0907.4/14262) for field work.

The following information was supplied regarding the deposition of DNA sequences:

All newly amplified nucleotide sequences are available at GenBank: MT235929–MT235968 (COI), MT248221–MT248260 (16S), and MT248181-MT248220 (18S).

The following information was supplied regarding data availability:

The raw data are available in the Supplemental Files.

The following information was supplied regarding the registration of a newly described species:

Publication LSID: urn:lsid:zoobank.org:pub:F94C18CF-8E07-4D4B-94ED-4153854B237E

Macrobrachium naiyanetri LSID: urn:lsid:zoobank.org:act:22EBCA17-2E29-4193-9D9E-87CABCD65D7D

Macrobrachium palmopilosum LSID: urn:lsid:zoobank.org:act:8065628A-4EDF-49EF-BA5D-91588F53D284

Macrobrachium puberimanus LSID: urn:lsid:zoobank.org:act:EE26BC6C-07F6-4C94-8B80-6F736B11F91A

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
