# Peer review of "Molecular phylogeny and species delimitation of the freshwater prawn Macrobrachium pilimanus species group, with descriptions of three new species from Thailand"

_PeerJ, doi:10.7717/peerj.10137_

## Round 0.1 · original submission · Major Revisions

All three reviewers have concerns with your manuscript, in several areas, including the evidence presented on morphology of your specimens, the failure to sufficiently cover additional sequence data available in other sources (e.g. BOLD) and there are also general concerns about the quality of the English in your manuscript, which could be improved. Please address all of the comments in your revisions.

·

Basic reporting

- The language was mostly clear, using professional standard English throughout. Only some parts required clarification for a better understanding of the scientific results, and I could spot and corrected a few typos, in particular in the Discussion part (see annotated pdf).

- The background and context were clearly detailed and the relevant existing litterature was correctly and sufficiently cited.

- The article is overall well structured. The tables were clear and the resolution of the figures was sufficient. All the relevant raw data were provided.

- This submission includes all the results relevant to the hypothesis.

Experimental design

- The results presented are original and agree with the Aims and Scope of the journal.

- The research question is well defined and the gap the results are meant to fill is clearly identified.

- The methods used to answer the question are the standard approach and were followed rigorously by the authors.

- Almost all the details needed to replicate the results were provided in this manuscript, a few parameters were missing in order to be able to reproduce the phylogenetic analyses (see annotated pdf).

Validity of the findings

No comment on the results, the evidence to support them is sufficient and state-of-the-art, and further research on the subject is encouraged by the authors.

Additional comments

This article presents a molecular phylogeny of the Macrobrachium pilimanus species group from Thailand with the descriptions of three new species. It is well structured and the presented results are sound and original. I could only find a few minor details that needed to be corrected, which is why I recommend this article to be accepted after minor revisions (see annotated pdf attached).

Reviewer 2 ·

Basic reporting

English needs to be improved. Unnecessarily verbose in places and wording often clumsy.

Choice of genetic markers need to be justified with reference to appropriate literature.

A brief discussion of species concepts is required as it is not clear how criteria were applied.

What is the "partitioned DNA sequence analysis" approach to species delimitation? No explanation or reference given.

Distributional information needs to be given, including sampling points and the know distribution of all species both within Thailand and the wider region. Map needs to include drainage basins.

Morphological data not presented in a way that demonstrated clear morphological differences among species. Cells in Table 4 could be highlighted for traits that are diagnostic for a particular species. Or a separate table presented giving the diagnostics traits. Even better would be a numerical taxonomic analysis in the form of a scatter diagram using multidimensional scaling or using a clustering algorithm or similar.

Too often we are asked to take on faith that groupings made by a taxonomists using morphology are valid without clearly presented evidence.

Experimental design

The stated aim is to "integrate traditional taxonomic examination and
molecular phylogeny using three molecular markers to delimit species boundaries"

However, the semi-subjective morphological-based assessments seem to drive the taxonomic conclusions and the molecular data is essentially provided as accessory information and is not critically evaluated. A better approach would be to demonstrate that all the designated taxa are morphological distinct and then look for support from the molecular data. Or if the morphological data is too variable, the paper should be driven by the molecular data.

A major gap in the data analysis is the failure to include relevant data from NCBI and BOLD for the same or related species of Macrobrachium. Figure S2 makes some effort, however this sample set is incomplete (not all samples from Figures1/2 are included). Further, Fig S2 indicates there are major issues with either species boundaries or identification in this group of species outside of Thailand, which requires further discussion.

18S must be dropped from the analysis - it is too conserved and often has erratic evolutionary rates between lineages. The species delimitation results for 18S in Figure 2, are all over the place.

The authors do not attempt to reconcile the differences between the range of species delimitation methods and gene regions. An MDS plot of genetic differences/similarity is often a useful way to determine/show where there are clusters of genetically similar samples that are divergent from other clusters.

Validity of the findings

There is frequently a problem with a country focused study of this kind when some of the species may occur widely outside the country. Thus it is essential that distributional information is included as the reader can not tell to what extent species are sampled from their full distributional range. Knowing a species is genetically distinct in sympatry is critical information, as well as what the distributional relationship is between species.

There is no attempt to reconcile the different outcomes of the species delimitation methods and gene regions. Clade B comes out as anywhere from a single species to up to 5 species (Figure 2), so there needs to be a rational basis for deciding how to delimit species. I don't to see any real attempt in "integration of traditional taxonomic examination and molecular phylogeny using three molecular markers to delimit species boundaries".

It is also necessary to validate the results using sequences on NCBI and BOLD. Figure S2 is step in the right direction, but this needs to be done for all related COI and 16S sequences. A better tabulation of within and between species genetic distance matrices is needed, and comparison with other Macrobrachium studies with respect divergence levels. While I don't advocate a purely bar-coding (distance) approach to species delimitation, there is plenty of discussion of the "bar-coding gap", which would be relevant here.

Additional comments

There are some good aspects to this paper including the attempted integration of tradition taxonomy and molecular genetic data (coupled with the use of species delimitation tools), however there are some serious limitations to how this has been done (or not done).

With significant revision this work maybe publishable. The major limitation will be trying to draw taxonomic conclusions for species that may have extensive distributions outside of Thailand. If the 3 putative new species can be justified and are known to be endemic to Thailand that would make it simpler. This why it is essential that the authors place their work in a clear geographic context.

·

Basic reporting

This paper is a nice blend of traditional morphology taxonomy and molecular taxonomy.

Experimental design

This paper is clearly written, largely well referenced, professionally presented.

Validity of the findings

I am no expert on morphological taxonomy, but it appears to have been well done to me.

The DNA barcoding (molecular taxonomy) seems reasonably well done, but there should be some discussion of the differing results from the different species delineation methods.

The phylogenetic element of the paper is the main bit that could use some further work.

The authors state that their results (line 256-258) “indicated the monophyletic relationship of all Thai species in the “pilimanus” species group (Clade A in 258 Figure 1) after being rooted by four congener species from other species groups”. Of course they are monophyletic if only 4 other taxa are included in the analysis from different lineages. Almost anything can be made monophyletic this way. To test monophyly, multiple exemplars of putative ingroup and putative outgroups must be included to make for a proper test.

There are plenty of available sequences on GenBank that could be used as a starting point. These include species supposedly within the pilimanus group from various countries (all of having at least COI or 16S sequences, and many with 18S too). These include: Macrobrachium pilimanus, Macrobrachium aff. pilimanus, Macrobrachium lepidactylus, Macrobrachium hirsutimanus, Macrobrachium dienbienphuense, Macrobrachium eriocheirum, Macrobrachium forcipatum, Macrobrachium platycheles, Macrobrachium amplimanus , Macrobrachium urayang. These need to be integrated, analysed and assessed. This can be done for COI, 16S & 18S separately, and then for all 3 together (Bayesian analysis can work well including some taxa with missing data).

Obviously, non-pilimanus group taxa need to be included too to test the monophyly of pilimanus, ideally taxa thought to be closely related, and more distant ones too. I note there is very little discussion of the results of previous papers in the setting of hypotheses/expectations for the current paper. For example there is Wowor et al 2009 MPE and Bracken et al 2010 Zool Scripta (I have not done an exhaustive literature search, there may well be many other relevant papers). I note the results of the current paper are different from Wowor et al 2009 in relation to the position of M. malayanum and M. niphane relative to pilimanus. This is the sort of detail that should be raised in the Introduction and then compared and discussed later.

So this sort of data and discussion needs to be included if the authors wish to speculate on larger scale phylogenetic relationships. The other option is to remove any real discussion of phylogeny, and just stick to morphological and molecular taxonomy. This would mean the authors would discuss the species that they have analysed but not deal much with higher level matters.

Additional comments

- A map is required as most readers won't know where these places are or how they relate to each other.

- The order of Table 1 is confusing. I would put it in alphabetical order by species, and within each species I’d organise it geographically (maybe by Thai regions of river basins). This would make it user friendly. It also needs to be clear that every site (except one) is in Thailand.

- The trees would be easier to interpret if there was some geographical labelling. Do the species group geographically? Within a species, is there obvious phylogeographic structure? This isn’t possible to assess without geographic labels of some sort.

---

## Round 0.2 · accepted · Accept

You have responded diligently to reviewers' comments, and your paper has significantly benefited from language improvements.